# Solubility Temperature Dependence of Bio-Based Levulinic Acid, Furfural, and Hydroxymethylfurfural in Water, Nonpolar, Polar Aprotic and Protic Solvents

Ana Jakob [1], Miha Grilc [1], Janvit Teržan [1] and Blaž Likozar [1,2,3,*]

1 Department of Catalysis and Chemical Reaction Engineering, National Institute of Chemistry, Hajdrihova 19, 1000 Ljubljana, Slovenia; ana.jakob@ki.si (A.J.); miha.grilc@ki.si (M.G.); janvit.terzan@ki.si (J.T.)
2 Pulp and Paper Institute, Bogišićeva 8, 1000 Ljubljana, Slovenia
3 Faculty of Polymer Technology, Ozare 19, 2380 Slovenj Gradec, Slovenia
* Correspondence: blaz.likozar@ki.si; Tel.: +386-1-476-0281

**Abstract:** Bio-based levulinic acid (LA), furfural (FF), and hydroxymethylfurfural (HMF) represent key chemical intermediates when biorefining biomass resources, i.e., either cellulose, glucose, hexoses, etc. (HMF/LA), or hemicellulose, xylose, and pentose (FF). Despite their importance, their online in situ detection by process analytical technologies (PATs), solubility, and its temperature dependence are seldom available. Herein, we report their solubility and temperature dependence by examining n-hexane, cyclohexane, benzene, toluene, 1,4-dioxane, diethyl ether, dichloromethane, tetrahydrofuran, ethyl acetate, acetone, dimethylformamide, acetonitrile, dimethyl sulfoxide, formic acid, n-butanol, n-propanol, ethanol, methanol, and water. These solvents were selected as they are the most common nonpolar, polar aprotic, and polar protic solvents. Fourier-transform infrared (FTIR) spectroscopy was applied as a fast, accurate, and sensitive method to the examined solutions or mixtures. The latter also enables operando monitoring of the investigated compounds in pressurized reactors. Selected temperatures investigated were chosen, as they are within typical operating ranges. The calculated thermodynamic data are vital for designing biorefinery process intensification, e.g., reaction yield optimization by selective compound extraction. In addition to extracting, upstream or downstream unit operations that can benefit from the results include dissolution, crystallization, and precipitation.

**Keywords:** furfural; hydroxymethylfurfural; levulinic acid; solubility; FTIR

## 1. Introduction

In recent years, environmental pollution and an increased need for energy have stimulated the research of biomass-based products as an alternative to the petroleum-based products. Production of bio-based fuels is a rapidly expanding field that promotes the investigation of chemicals with promising potential in this field. Biofuels and bio-based chemicals can be produced from renewable sources, such as biomass [1–7]. This could potentially lead to an efficient management of greenhouse gas emissions and a circular economy. Lignocellulosic biomass resources, composed of cellulose, hemicellulose, and lignin, represent one of the most viable options to produce biofuels, bioadditives, and biochemicals, especially as they do not directly compete with agricultural food crops, as does vegetable oil biodiesel. Nonetheless, as opposed to fossil resource refining, biorefining suffers from difficult operation optimization, as the amount, composition, and impurities of the feedstock change over time. Biorefineries are, almost as a rule, operated intuitively as well as suboptimally. Hydroxymethylfurfural, bio-based levulinic acid, and furfural are the main reaction products when decomposing (hemi)cellulose [8–13]. They can be formed by the dehydration of sugars [8,11,14–16]. Moreover, they are also the most notable intermediates for biofuels, that is, via hydrogenation, esterification or aldol condensation mechanisms. One of the crucial aspects in these production process, as in all production

processes, is purity. Optimizing these noted conversions is difficult due to the changes in resources themselves, which calls for the implementation of fast online measurements that serve as a basis for the variation of operating process conditions that consider feedstocks. Given that solvents also vary along the biorefining operation up- or downstream, the concentrations in different solvents should be considered, ranging from the aqueous biomass pretreatment to the (solvent-based) reaction/extraction unit operations, which is inherent to furans. Solubility is one of the key factors that determine production efficiency. It is especially important during the extraction and other processes such as crystallization, distillation, and chromatography, which are frequently used as purification methods. Consequently, it is reasonable to focus on solubility due to its importance in the production process and also due to the fact that the available data on this parameter are very limited [7]. Determining the concentration can be difficult, especially quickly and precisely. Several very accurate techniques have been developed [17,18], but they are not easily used on-line.

Furfural (FF, 2-furaldehyde) is an aromatic aldehyde with the molecular formula $C_5H_4O_2$; it has a melting point at $-38.1\ ^\circ C$, a boiling point at $161.7\ ^\circ C$, and is a liquid at room temperature [11]. It has limited solubility in water, but it dissolves well in organic solvents such as DMSO, ethanol, acetone, chloroform, diethyl ether, tetrahydrofuran (THF), and benzene [19,20]. There is limited information available about the temperature dependence of solubility, especially for organic solvents. On the other hand, water solubility is known in the temperature range of 10–100 $^\circ C$ [21]. These measurements are reported without pH adjustments [19,21–23]. Furthermore, furfural itself is also used as a solvent in some reactions due to its physical–chemical characteristics. It was excellently described by Eseyin and Steele as follows: "Furfural is commonly used as a solvent; it is soluble in ethanol and ether and somewhat soluble in water. The aldehyde group and furan ring in furfural confers the furfural molecule with outstanding properties as a selective solvent" [24].

Hydroxymethylfurfural (HMF), 5-(hydroxymethyl)furan-2-carbaldehyde, with the molecular formula $C_6H_6O_3$, has a melting point at 28–35 $^\circ C$ and the boiling point at 114–116 $^\circ C$ (1 mbar), and at room temperature it is a solid [11]. HMF has excellent solubility in water, and it exhibits the same property in some organic solvents, such as dimethyl sulfoxide (DMSO) and tetrahydrofuran (THF). According to available data, a wide range of substances, namely GVL, MeOH, BuOH, n-hexane, n-decane, n-decene, toluene, MIBK, SADE, EtOAc, BuOAc, 1,4-DO, DBE, MTBE, were mentioned as potential solvents. The measurements were performed only at two temperatures, 25 and 30 $^\circ C$, with no pH adjustments included, and so the temperature and pH dependence of solubility in this research was not reported [25,26].

Both HMF and FF are precursors of levulinic acid (LA), the third discussed derivate in this study. Levulinic acid (LA), 4-oxopentanoic acid, with the molecular formula $C_5H_8O_3$ has a melting point at 33 $^\circ C$ and a boiling point at 245–246 $^\circ C$ [9,11]. LA is soluble in water and in different organic solvents, more precisely, methanol, ethanol, chloroform, diethyl ether, acetonitrile, ethyl lactate, ethyl acetate, and toluene. Research showed that LA is insoluble in nonpolar isooctane and cyclohexane. [27–29] The available data only reports the solubility in water in the temperature range of 5–31 $^\circ C$ and pH of 4.7–9.6 [21,27,28].

In the end, there is some data available on water solubility of FF, LA, and HMF, with a few including temperature and pH dependence [23,26,28]. The data concerning the solubility in organic solvents are even scarcer. Therefore, the aim of this work is to deepen the available knowledge of solubility for all three substances in a wide range of solvents at different concentrations and temperatures. The main goal of this research is to determine the solubility of FF, LA, and HMF in order to find potential solvents for production processes to ease downstream purification, such as crystallization. Additionally, knowledge of solubility is expected to aid the development of new catalytic routes as well as enhance already established ones. In many cases, the solvent not only affects the reactants but also the catalysts [30,31].

In this work we describe the use of FTIR as a fast, accurate, and sensitive method for measuring the concentration of the abovementioned compounds. The use of FTIR peak height as the concentration determining factor has already been established for use with hydrocarbons [32,33]. More specifically, the use of the carbonyl group peak for the determination of concentration has already been described in the literature [34]. To support the validity of the reported data, we describe the procedure in detail as was excellently described and suggested by Königsberger [35].

## 2. Materials and Methods

### 2.1. Materials

The chemicals used in this study, including their source, purity, and melting point stated by the supplier are presented in Table 1. Additional information about the solutes and solvents can be found in Table 2 and Table S1.

**Table 1.** Chemicals, source, purity, CAS number, and melting point ($T_m$).

| Name | Purity | Source | CAS Number | $T_m$ [°C] |
|---|---|---|---|---|
| Levulinic acid | 98% | Sigma-Aldrich | 123-76-2 | 30–33 |
| Furfural | 99% | Sigma-Aldrich | 98-01-1 | −36 |
| Hydroxymethylfurfural | | | 67-47-0 | 28–33 |
| n-Hexane | 98.5% | Merck | 110-54-3 | −94 |
| Cyclohexane | 99.5% | Elixir Zorka Šabac | 110-82-7 | 6.5 |
| Benzene | 99.0% | Honeywell | 71-43-2 | 5.5 |
| Toluene | 99.7% | Honeywell | 108-88-3 | −95.0 |
| 1,4-dioxane | 99% | Honeywell | 123-91-1 | 12 |
| Diethyl ether | 99.5% | Merck | 60-29-7 | −116.3 |
| Dichloromethane | 99.5% | Merck | 75-09-2 | −95 |
| THF | 99% | Sigma-Aldrich | 96-47-9 | −108 |
| Ethyl acetate (ETOAc) | 99.9% | Honeywell | 141-78-6 | −84 |
| Acetone | 99.5% | Honeywell | 67-64-1 | −95 |
| Acetonitrile | 99.9% | Merck | 75-05-8 | −45.7 |
| DMSO | 99.9% | Merck | 67-68-5 | 18.5 |
| Formic acid | 100% | Merck | 64-18-6 | 4 |
| n-Butanol | 99.5% | Merck | 71-36-3 | −89 |
| n-Propanol | | Alkaloid Skopje | 71-23-8 | −126.0 |
| Ethanol (EtOH) | 99.9% | Carlo Erba | 64-17-5 | −114.1 |
| Methanol (MeOH) | 99.9% | Honeywell | 67-56-1 | −98 |
| $H_2O$ | | Distilled semi-Q | | 0.0 |

**Table 2.** The solvents used, grouped by polarity and proticity.

| Nonpolar | Polar | |
|---|---|---|
| | Aprotic | Protic |
| n-Hexane | Dichloromethane | Formic acid |
| Cyclohexane | THF | n-Butanol |
| Benzene | Ethyl acetate | n-Propanol |
| Toluene | Acetone | Ethanol |
| 1,4-Dioxane | Acetonitrile | Methanol |
| Diethyl ether | DMSO | $H_2O$ |

### 2.2. Experiment Procedure

Firstly, 18 solvents were selected to assess the qualitative solubility of all three chemicals, i.e., FF, LA, and HMF. Samples with a concentration of 20 g L$^{-1}$ were prepared in a 1 mL vial and after 3 min of hand stirring, the solubility was visually estimated. This was a prescreening method to determine whether the solvent was appropriate. The method was suitable for FF and HMF since they are an orange-brown color. This was not possible for colorless LA, which happened to be a liquid at 30–33 °C. The solvents (Table 1), which

dissolved but did not react with a selected chemical after stirring, were further examined with an on-line FTIR method. The only case where the solvent interacted with the solutes was in the case of formic acid, where the color green was seen immediately after mixing.

Aliquots of 3 mL were prepared in 5 mL vials for all three investigated compounds and for pure solvents. There were 4 aliquots prepared for each of the former of 4 different concentrations (20, 50, 100, and 200 g L$^{-1}$) and analyzed with on-line FTIR spectroscopy. More details on the experimental procedure are available in the Supplementary Materials. The first spectrum of pure solvent was collected, and then samples of four different concentrations were analyzed. From the measurements of the mixtures, the spectrum of pure solvent was subtracted. The spectra were collected until the absorbance was stable and until we were certain equilibrium was reached. In most cases the spectra were collected for 100 s (Figure S1). Afterwards, the samples and the pure solvent were placed into the double jacketed glass reactor filled with EtOH (96.0%, ECP) and antifreeze (in the jacket) to cool the samples to 0 °C (±0.5) and then to −10 °C (±0.5). Samples of FF and HMF were protected from visible light with aluminum foil, due to their tendency to degrade. They were also not prepared in advance, since FF is noticeably hygroscopic. The experimental procedure was the same for all three tested chemicals.

Some solvents (cyclohexane, benzene, 1,4-dioxane, DMSO, and H$_2$O), were impossible to analyze at all three temperatures, due to their higher melting points.

### 2.3. Analytical Method

Samples were analyzed with the ReactIR 45 m in situ FTIR spectroscopy (Mettler Toledo) equipped with a diamond probe and with iC IR software. We scanned the spectrum in a range of 650–2000 cm$^{-1}$. The samples were analyzed at three different temperatures, i.e., room temperature (from 24 to 26 °C), 0 °C, and −10 °C. The measuring surface was cleaned between scans with ethanol (96.0%, ECP). Spectra were collected every 15 s, and each measurement represents an average of 50 scans.

## 3. Results

### 3.1. FTIR Spectrum of Furfural, Levulinic Acid, and Hydroxymethylfurfural

A summary of most relevant FTIR spectra vibrations for each individual compound are listed in Table 3. The spectrum of furfural was measured at room temperature. The IR spectrum (Figure 1) shows a very strong absorption peak in the area around 1650–1700 cm$^{-1}$. These two peaks at wavenumbers 1671 and 1696 cm$^{-1}$ belong to the conjugated aldehyde carbonyl group (–H–C=O). The overall absorption intensity was slightly lower than usual, due to the internal formation of hydrogen bonds, which usually occurs in conjugated unsaturated aldehydes. As we added the solvent, a division of these two specific peaks became more noticeable and could be explained with isomerization of the aforementioned aldehyde group. The isomerization or so-called Fermi resonance can be also noticed at the area around 1365–1395 cm$^{-1}$, representing the C–H bond in the aldehyde group. In our case, this corresponds with slightly shifted peaks at 1369 and 1395 cm$^{-1}$, belonging to cis and trans isomers of furfural [36] (Figure 1a). Peaks at 1466 and 1570 cm$^{-1}$ stand for two C=C bonds as part of an aromatic ring in the FF structure. A C–O vibration is represented by two peaks at 1156 and 1279 cm$^{-1}$. Vibrations assigned to C–H and ring deformations were observed at 1018 and 1081 cm$^{-1}$. The out-of-plane =C–H bond or sp$^2$ hybridized C–H show absorption in the region from 750–929 cm$^{-1}$ [37] (Figure 1a).

The IR spectrum of LA shows strong and broad signals at 1704 cm$^{-1}$ and 1742 cm$^{-1}$ that correspond with both C=O functional groups in molecule. The CH$_3$ symmetric and asymmetric bending vibrations were noticed at 1369 and 1402 cm$^{-1}$. The peak at 1163 cm$^{-1}$ can be attributed to C–O acid stretch vibration [38] (Figure 1b).

HMF and LA were also measured at a higher temperature (40 °C), since they have higher melting points than furfural. This was then compared to the measurements of solids done at 25 °C, to see whether any changes in the spectra occur during the phase change. No changes were observed. The IR spectrum of HMF shows a very strong absorption

at wavenumber 1663 cm$^{-1}$, indicating that it belongs to the carbonyl group (C=O). The series of peaks from 1521 to 1342 cm$^{-1}$ is attributed to C=C stretches. At 1193 cm$^{-1}$ the =C–O–C= stretch was detected. As with furfural, the in-plane C–H deformation and the ring deformation at 1018–1070 cm$^{-1}$ were also noticed. The two peaks (Figure 1c) at 772–809 cm$^{-1}$ are indicative of out-of-plane C–H deformation [38].

**Table 3.** Summary of relevant FTIR spectra vibrations.

| Frequency (cm$^{-1}$) | Band Assignment | Species |
|---|---|---|
| 929 | C−H out-of-plane deformation | Furfural |
| 1018 1081 | Ring deformation | |
| 1156 1279 | =C–O–C= ring vibration | |
| 1570 | C=C stretches | |
| 1696 1671 | C=O aldehyde stretch | |
| 1163 | C−O acid stretch vibration | Levulinic acid |
| 1369 1402 | CH$_3$ symmetric and asymmetric bending vibration | |
| 1704 1742 | C=O aldehyde stretch | |
| 772−809 | C−H out of plane deformation | HMF |
| 1018 1070 | Ring deformation | |
| 1193 | =C–O–C= ring vibration | |
| 13,421−521 | C=C stretches | |
| 1663 | C=O aldehyde stretch | |

The spectra of all three samples (FF, HMF and LA) that were dissolved in acetone, acetic acid, and ethyl acetate were as expected, with no abnormal or unexpected vibrations. A broad peak at approximately 1700 cm$^{-1}$ (from the carbonyl group of the mentioned solvents) overlapped with the peaks of all three samples. All the experimental data are presented in the supplementary information file (Tables S2–S7).

### 3.2. Concentration Dependent Solubility

The concentration dependence of the IR signal was studied (Figure 2). Since every single one of the studied molecules has at least one carbonyl group with a very strong absorption peak in all the obtained spectra, the latter was chosen as the observation point for solubility. In all solutions, the specific wavenumber of the carbonyl peak belonging to each compound was selected. In an acetonitrile solution, the observed wavenumbers were 1696, 1720, and 1673 cm$^{-1}$, for FF, LA, and HMF, respectively (Figure 2a,d). Additional FTIR graphs can be found in the Supplementary Materials Figures S2–S4. In most cases, the linear increase in concentration was accompanied by a linear increase in peak intensity. The results were used to prepare a calibration curve and to determine the approximate concentration of solubility up to 200 g L$^{-1}$ of the added solute. If the response is linear, we can assume very good solubility and that we are not at the maximum solubility yet. For FF, we see a linear response with $R^2 > 0.990$ in THF and diethyl ether solution at 25 °C. For LA, this was observed in acetone and acetonitrile, as well as THF, all at 25 and 0 °C. HMF was found to be very well soluble in 1,4-dioxane, acetonitrile, acetone, THF, and in MeOH at 25 °C (Table 4; Figure 2d–f).

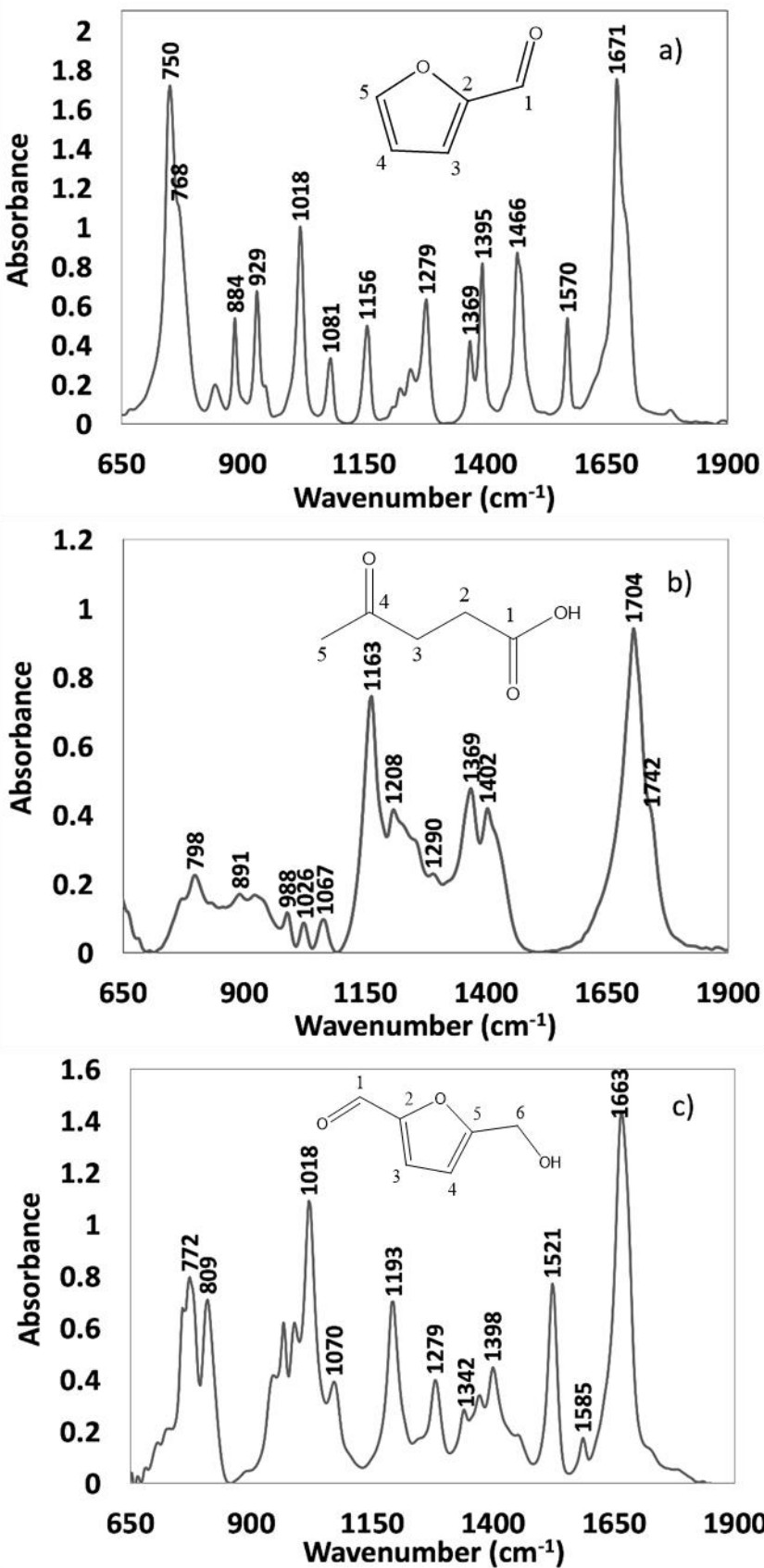

**Figure 1.** FTIR spectrum at 25 °C and the structural formula of (**a**) furfural, (**b**) levulinic acid, and (**c**) hydroxymethylfurfural.

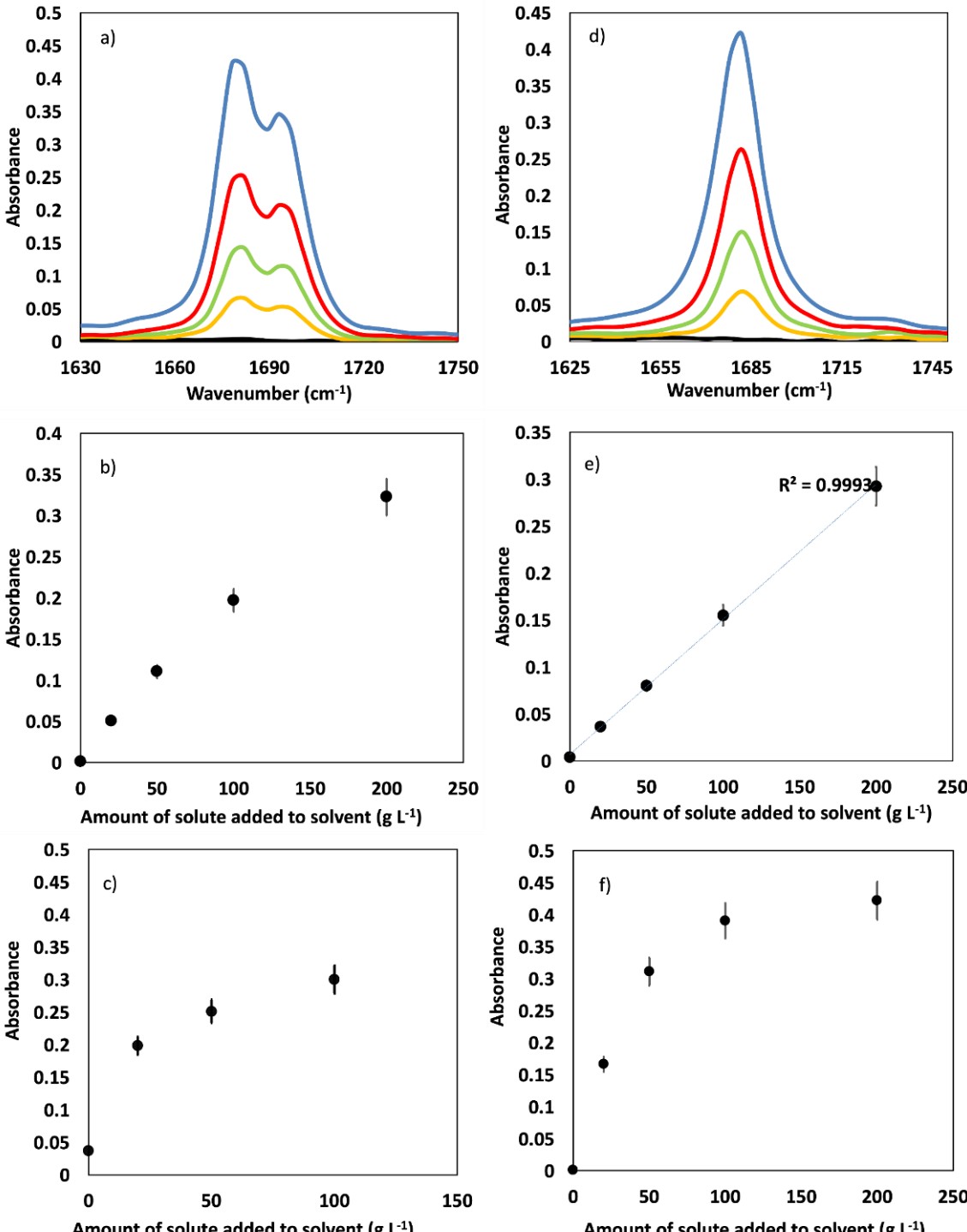

**Figure 2.** (**a**) Absorbance of FF in acetonitrile at 25 °C (black for acetonitrile, yellow for 20 g L$^{-1}$, green for 50 g L$^{-1}$, red for 100 g L$^{-1}$, blue for 200 g L$^{-1}$); (**b**) signal intensity at 1696 cm$^{-1}$ (the carbonyl group) of FF in acetonitrile; (**c**) signal intensity of FF at 1696 cm$^{-1}$ in water (nonlinear response); (**d**) absorbance of HMF in acetonitrile; (**e**) signal intensity at 1673 cm$^{-1}$ of HMF in acetonitrile (linear response, R$^2$ = 0.9993); (**f**) signal intensity at 1673 cm$^{-1}$ of HMF in water (nonlinear response).

**Table 4.** Calculated values of solubility for the three examined compounds in selected solvents.

| Solvent | Calculated Solubility of Furfural in 15 Different Solvents in g L$^{-1}$ | | | Calculated Solubility of Levulinic Acid in 14 Different Solvents in g L$^{-1}$ | | | Calculated Solubility of Hydroxymethylfurfural in 12 Different Solvents in g L$^{-1}$ | | |
|---|---|---|---|---|---|---|---|---|---|
| | −10 °C | 0 °C | 25 °C | −10 °C | 0 °C | 25 °C | −10 °C | 0 °C | 25 °C |
| n-Hexane | IN | IN | IN | IN | IN | IN | IN | IN | IN |
| Cyclohexane | IN | IN | IN | IN | IN | IN | IN | IN | IN |
| Benzene | * | * | 570 ± 40 | * | * | 430 ± 30 | IN | IN | IN |
| Toluene | 430 ± 30 | 480 ± 30 | 610 ± 40 | 250 ± 20 | 450 ± 30 | 470 ± 30 | IN | IN | IN |
| 1,4-Dioxane | * | * | 1110 ± 80 | * | * | 1020 ± 70 | * | * | FM |
| Diethyl ether | 420 ± 30 | 1040 ± 70 | FM | 620 ± 40 | 620 ± 40 | 620 ± 40 | IN | IN | IN |
| Dichloromethane | 550 ± 40 | 670 ± 50 | 870 ± 60 | 500 ± 40 | 520 ± 40 | 570 ± 40 | 560 ± 40 | 590 ± 40 | 700 ± 50 |
| THF | 740 ± 50 | 760 ± 50 | FM | 940 ± 70 | FM | FM | 560 ± 40 | FM | FM |
| EtOAc | 850 ± 60 | 860 ± 60 | 1030 ± 70 | 400 ± 30 | 690 ± 50 | 850 ± 60 | 660 ± 50 | 660 ± 50 | 660 ± 50 |
| Acetone | 760 ± 50 | 810 ± 60 | 810 ± 60 | FM | FM | FM | FM | FM | FM |
| Acetonitrile | 910 ± 60 | 910 ± 60 | 910 ± 60 | FM | FM | FM | FM | FM | FM |
| DMSO | * | * | 820 ± 60 | * | * | 610 ± 40 | * | * | 660 ± 50 |
| Formic acid | DG | DG | DG | DG | DG | DG | DG | DG | DG |
| n-Butanol | 760 ± 50 | 810 ± 60 | 850 ± 60 | 840 ± 60 | 840 ± 60 | 840 ± 60 | 920 ± 60 | 920 ± 60 | 920 ± 60 |
| n-Propanol | 720 ± 50 | 810 ± 60 | 820 ± 60 | 610 ± 40 | 730 ± 50 | 800 ± 60 | 530 ± 40 | 530 ± 40 | 720 ± 50 |
| EtOH | 760 ± 50 | 840 ± 60 | 990 ± 70 | FM | 480 ± 30 | 930 ± 70 | 670 ± 50 | 800 ± 60 | 1030 ± 70 |
| MeOH | 690 ± 50 | 720 ± 50 | 960 ± 70 | 890 ± 60 | 890 ± 60 | 890 ± 60 | 580 ± 40 | 1020 ± 70 | FM |
| H$_2$O | * | * | 90 ± 10 | * | * | NS | * | * | 180 ± 10 |

IN—insoluble at 20 g L$^{-1}$ and above, *—no spectra, due to higher melting point, NS—no spectra, due to unstable signal, DG—degradation of solute, FM—fully miscible, g L$^{-1}$ = amount of solute added to solvent. Grey—not applicable (NA), red—insoluble (IN), orange—moderate solubility (MS), yellow—good solubility (GS), green—excellent solubility (ES).

FF formed a two-phase system with water and hexane at 100 g L$^{-1}$ of the added solute and above, which is indicative of poor miscibility. The obtained results for the immiscibility of FF in water and hexane were also reported by other researchers [39]. The solubility of FF in these two solvents is represented by the formation of a plateau at a concentration of 100 g L$^{-1}$ in the solubility curve (Figure 2c). The FTIR of the aqueous solutions of the other two compounds (HMF and LA) was unstable, which can be attributed to very strong hydrogen bonding with the solvent. FF was insoluble in pentane and cyclohexane, and it degraded in formic acid (Table 4). LA was insoluble in pentane, hexane, and cyclohexane; it also degraded in formic acid (Table 4). HMF has poor solubility in the majority of the examined solvents; it already reached maximum solubility at 20 g L$^{-1}$ of the tested solvents, and HMF tended to form a visible suspension (Table 4).

A common trend in absorbance was noticed for all three compounds. Generally, the absorbance was lower in protic polar solvents and higher in nonpolar solvents. An exception was DMSO, where the recorded absorbance was the highest for all of three solutes.

### 3.3. Signal Dependence at Different Temperatures

FF and LA have a tendency to crystallize in toluene at lower temperatures. Toluene, however, was the only solvent in which FF and LA crystallized. For FF, the crystallization initiates when the solution is cooled to −10 °C. The crystals were transparent at lower concentrations and turned a brownish color at higher concentrations (200 g L$^{-1}$). When the sample was heated up to 35 °C, the crystals dissolved, demonstrating that the crystallization of FF in toluene is reversible. The crystallization of FF was also noticeable in the intensity of the IR signal, which was lower at −10 °C (Figure 3b). HMF also has a tendency to crystalize in toluene, but we found that it is insoluble at 25 °C [40,41].

When dissolving LA in toluene at −10 °C, we noticed that when the amount of added solute per solvent reached 50 g L$^{-1}$, transparent samples became opaque. We assume this to be the crystallization of LA due to the global supersaturation of the solution (Figure 3a). This phenomenon caused a drastic jump in absorbance in the entire FTIR spectrum. The crystals of LA floating in the medium caused the IR beam to be scattered, which in turn caused the jump in absorbance. Crystallization of LA was also observed by other researchers [42] at −15 °C, which correlates with our findings.

As expected, some solvents (benzene, 1,4-dioxane, and DMSO), were impossible to analyze at lower temperatures (0 and −10 °C) due to their higher melting points. Other trends related to temperature are discussed below with calculated solubility.

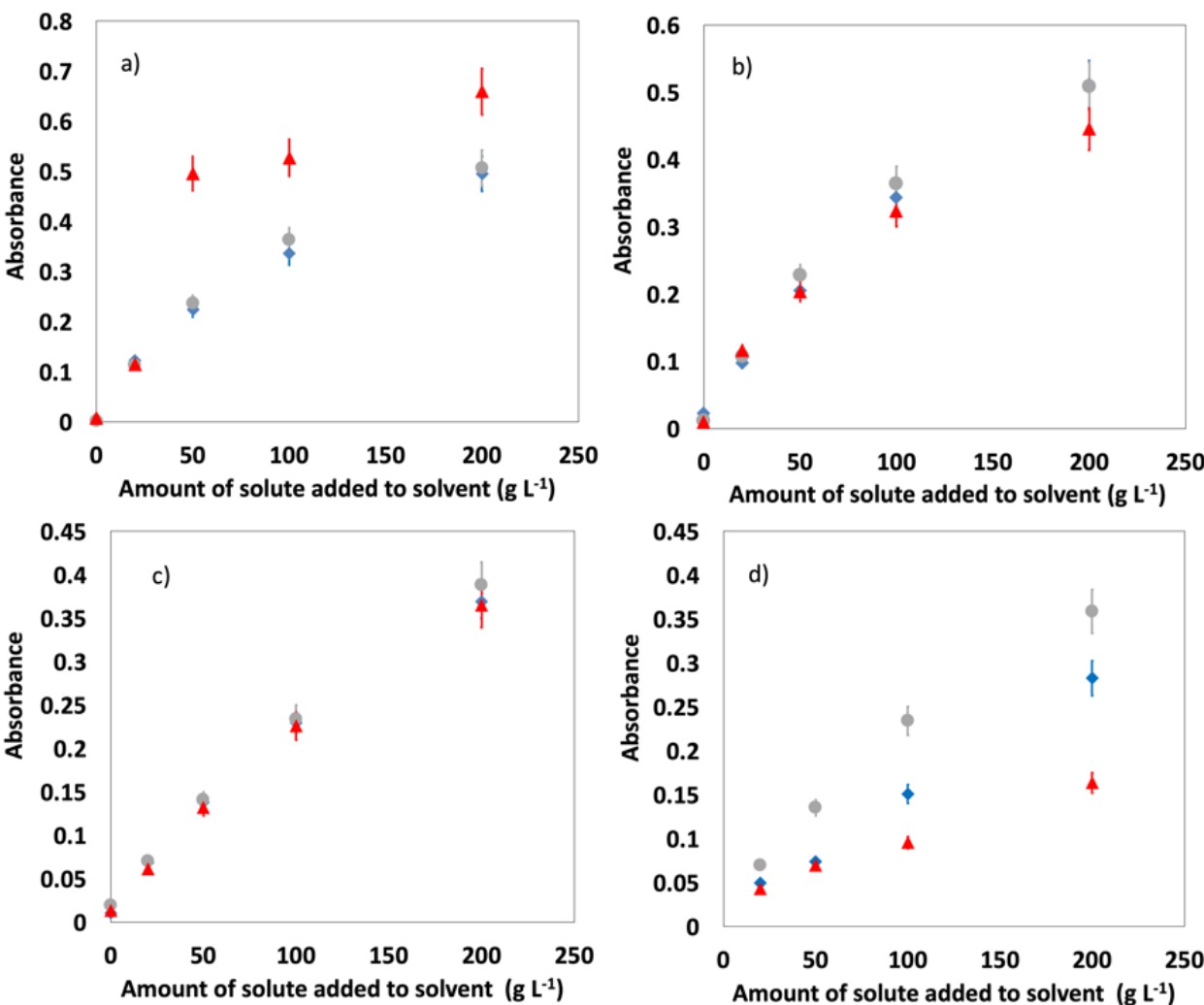

**Figure 3.** (**a**) intensity of the carbonyl peak in a LA-toluene solution at −10 °C, 0 °C, 25 °C; (**b**) intensity of the carbonyl peak in a FF-toluene solution, crystallization at −10 °C results in a lower signal intensity at −10 °C, 0 °C, 25 °C; (**c**) intensity of the carbonyl peak in a LA-n-butanol solution (no noticeable temperature dependence) at −10 °C, 0 °C, 25 °C; (**d**) intensity of the carbonyl peak in a HMF-EtOAc solution (significant temperature dependence). Red triangle indicates −10 °C; blue diamond, 0 °C; grey circle, 25 °C.

### 3.4. Solubility

For experimental measured data obtained with direct online FTIR spectroscopy that could not be fitted with a linear regression curve, an exponential fitting method with Equation (1) was applied.

$$\text{Absorbance} = A\,(1 - e^{(-Bc)}) \tag{1}$$

For Equation (1), where c (g L$^{-1}$) corresponds to concentration, the constants A (/) and B (L g$^{-1}$) were calculated for each individual experiment (Tables S8–S10) using linearization and linear regression. From the Equation (1), the concentration of solubility was calculated as the value at 99% maximal liquid absorbance.

Taking into account all the data obtained, the general conclusion is that FF has a good solubility in a variety of organic solvents [19]. It is insoluble in pentane and cyclohexane [43]. In hexane, a two-phase system was formed even at a 100 g L$^{-1}$ of added solute. The latter three solvents are highly nonpolar and are not able to form π–π interactions; consequently, this leads to the insolubility of FF. For other cyclic aromatic carbohydrates, benzene and toluene solubilities were calculated, and a temperature dependence was noticed. At lower temperatures, a lower concentration of the solute was expected. This trend also

occurred in solvents such as diethyl ether, dichloromethane, THF, ethyl acetate, n-butanol, n-propanol, methanol, and ethanol. We determined a good solubility of FF in 1,4-dioxane. In acetonitrile solutions, we did not notice a temperature dependence of solubility. The calculated solubility of furfural in water ($90 \pm 6$ g $L^{-1}$) matched well with the value reported in the literature (90 g $L^{-1}$). If the maximum solubility was exceeded, a two phase system formed, unless the solute and the solvent exhibited full miscibility [22]. The solubility of furfural in water was the lowest value calculated, and the reason most likely involves the highly polar character of water (Table S1) [19,21–23].

Similar to furfural, levulinic acid was insoluble in pentane, hexane, and cyclohexane. The solvents, in which solubility exhibited a temperature dependence, were toluene, dichloromethane, ethyl acetate, THF, and n-propanol. The drop in solubility was significant, especially when the temperature reached $-10\ ^\circ$C. In diethyl ether, acetone, acetonitrile, n-butanol, and methanol, solute concentration values were not temperature dependent. Really good solubility, with an almost linear response (IR signal increased linearly, $R^2 > 0.990$) was detected in acetone and acetonitrile solutions (Table 4) [27–29].

The polarity of hydroxymethylfurfural is higher when compared to FF, due to its extra hydroxymethyl functional group on the fifth C atom (Figure 1c). As a result, it is virtually insoluble in highly nonpolar solvents such as pentane, hexane, and cyclohexane. Unlike furfural, HMF is also insoluble in toluene, benzene, and diethyl ether, also attributable to the higher polarity. The latter additionally leads to higher solubility in water. The solute concentration increased two-fold compared to FF. In methanol, ethanol, n-propanol, THF, and dichloromethane, we noticed a stronger temperature dependence of solubility. Ethyl acetate, acetone, acetonitrile, and n-propanol solvents show almost no temperature dependence. For 1,4-dioxane, acetonitrile, and acetone, the signal response in FTIR indicates a good solubility (linear response, $R^2 > 0.990$). In all solvents, except for DMSO, THF, and dichloromethane, we can notice the formation of a suspension (Table 4) [25,26].

The calculated values of solubility are shown in Table 4 with a relative standard deviation of 7.0% (RSD = 7.0%). The error was calculated by looking at the difference in peak height between the spectra measured immediately after mixing and after the equilibrium formed. All the calculations were based on experiments, and they were additionally validated by GC-MS experiments (Figures S5–S7).

## 4. Discussion

The predicted solubility for furfural in water correlates very well with reported data [44]. The data on the solubility in benzene, ethanol, acetone, and ethyl ether also correlate with reports from the literature [45]. We could not compare the data for levulinic acid with reports from the literature, as we found, after an extensive search, only reports for water solubility. For hydroxymethylfurfural, the data that are available correlate with the data in this manuscript [46]. However, exact data for the solubility of all three compounds are still scarce and hard to come by.

## 5. Conclusions

In our work we successfully used FTIR spectroscopy to estimate the solubility of furfural, levulinic acid, and hydroxymethylfurfural in 18 different solvents. As all three compounds are important renewable feedstock chemicals, we believe a database, as presented in this study, is essential for further studies. We found that in some cases, such as furfural in toluene, there is a notable temperature dependence on solubility. In some cases, however, such as levulinic acid in diethyl ether, very little correlation was observed between solubility and temperature. Calculated thermodynamic data are vital for designing biorefinery unit operation.

**Supplementary Materials:** The supplementary file is available online at https://www.mdpi.com/article/10.3390/pr9060924/s1.

**Author Contributions:** Conceptualization, M.G. and B.L.; methodology, M.G. and B.L.; validation, A.J. and J.T.; investigation, A.J.; resources, M.G. and B.L.; data curation, A.J. and J.T.; writing—original draft preparation, A.J.; writing—review and editing, A.J., M.G., J.T., and B.L.; supervision, M.G. and B.L.; funding acquisition, M.G. and B.L. All authors have read and agreed to the published version of the manuscript.

**Funding:** This research was funded by the Slovenian Research Agency (research core funding P2-0152 and basic postdoctoral research project Z2-9200). The work was partially carried out within the RDI project Cel.Cycle, "Potential of biomass for development of advanced materials and biobased products", which is cofinanced by the Republic of Slovenia, Ministry of Education, Science and Sport, and the European Union through the European Regional Development Fund, 2016–2020.

**Conflicts of Interest:** The authors declare no conflict of interest.

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
