# Peer review of "Solubility Temperature Dependence of Bio-Based Levulinic Acid, Furfural, and Hydroxymethylfurfural in Water, Nonpolar, Polar Aprotic and Protic Solvents"

_processes, doi:10.3390/pr9060924_

Round 1
Reviewer 1 Report
Comments to the paper “Solubility Temperature Dependence of Bio-Based Levulinic 2 Acid, Furfural and Hydroxymethylfurfural in Water, Nonpolar, 3 Polar Aprotic and Protic Solvents” by Ana Jacob et al.
In the presented manuscript, Authors inventively used FTIR spectroscopy to estimate the solubility of three important platform compounds originating from lignocellulosic biomass, namely furfural, levulinic acid and hydroxymethylfurfural in 18 different solvents. They found good correlation with previously reported data that presently available in the literature.
These data can be useful for future development of separation processes and also seem to be essential for further studies. The paper is well written and concise, therefore I have only some minor comments:
- It would be useful to summarize the characteristic frequencies for different compounds in a table (see part 3.1.)
- Some text is missing at line 302.
- A few typos left in the text.
Author Response
1. It would be useful to summarize the characteristic frequencies for different compounds in a table (see part 3.1.)
According to your comment, we added a list characteristic frequencies for each compound in Table 3 on page 5.
2. Some text is missing at line 302.
Lines 303-305 were accidentally left in from a previous version and were correctly removed.
3. A few typos left in the text.
The manuscript was carefully checked for typos.

Reviewer 2 Report
The manuscript prepared by the team of Ana Jakob et al. Presents significant dissolution studies of bio-based levulinic acid, furfural, and hydroxymethylfurfural in water, nonpolar, polar aprotic, and protic solvents.
These compounds are important for the production of biofuels and green chemistry. It is worth emphasizing that the presented studies using the known FT-IR method present new data on the solubility of the tested compounds in several solvents.
Research is properly planned, carried out, and presented. The visible aspect of novelty and practical application.
My comments relate only to the editorial site:
- please check and disable the presence of double spaces, especially for units
- I would recommend changing the caption under Fig.1 "In this figure we see ..."
- In Fig. 1, part c), the letter c appears under the formula of the compound. What does it mean?
- In verse 287 it mentions: "extra hydroxymethyl functional group on the fifth C atom" - where is the numbering shown? Which atom is the fifth atom?
- Line 303-305 - template residue, please delete
Author Response
1. Please check and disable the presence of double spaces, especially for units
According to your comment, this was corrected.
2. I would recommend changing the caption under Fig.1 "In this figure we see ..."
The caption of Fig. 1 was corrected to “FT-IR spectrum at 25°C and the structural formula of: a) furfural, b) levulinic acid, c) hydroxymethylfurfural“.
3. In Fig. 1, part c), the letter c appears under the formula of the compound. What does it mean?
This c was a typo and was removed from the image.
4. In verse 287 it mentions: "extra hydroxymethyl functional group on the fifth C atom" - where is the numbering shown? Which atom is the fifth atom?
According to your comment we added the numbering of the molecules in the Figure 1 on page 7 and was referenced in the text (line 303).

Reviewer 3 Report
Work at a high level. I have no substantive reservations.
Please pay attention to the caption of Figure 1. Maybe it would be better: "Figure 1: FT-IR spectrum at 25°C and the structural formula of: a) furfural, b) levulinic acid, c) hydroxymethylfurfural"
In Figure 3, there are no commas between points a), b), c), d.
I have no more comments.
Author Response
1. Please pay attention to the caption of Figure 1. Maybe it would be better: "Figure 1: FT-IR spectrum at 25°C and the structural formula of: a) furfural, b) levulinic acid, c) hydroxymethylfurfural"
According to your comment we corrected the caption of Figure 1.
2. In Figure 3, there are no commas between points a), b), c), d.
As mentioned we corrected missing commas between points a), b), c) and d).
